# *Trichomonas vaginalis*: Monolayer and Cluster Formation—Ultrastructural Aspects Using High-Resolution Scanning Electron Microscopy

**DOI:** 10.3390/pathogens12121381

**Published:** 2023-11-23

**Authors:** Sharmila Fiama das Neves Ortiz, Raphael Verdan, Fabio da Silva de Azevedo Fortes, Marlene Benchimol

**Affiliations:** 1Laboratório de Ultraestrutura Celular Hertha Meyer, Instituto de Biofísica Carlos Chagas Filho, Centro de Pesquisa em Medicina de Precisão, Universidade Federal do Rio de Janeiro, Rio de Janeiro 21941-901, Brazil; sharmilaortiz@biof.ufrj.br (S.F.d.N.O.); raphaelverdan@biof.ufrj.br (R.V.); 2BIOTRANS-CAXIAS Campus, Universidade do Grande Rio, UNIGRANRIO, Rio de Janeiro 96200-000, Brazil; fabio.fortes.uerj@gmail.com; 3Laboratório de Terapia e Fisiologia Celular e Molecular, Departamento de Biologia, Faculdade de Ciências Biológicas e Saúde, Universidade do Estado do Rio de Janeiro, Rio de Janeiro 23070-200, Brazil; 4Instituto Nacional de Ciência e Tecnologia em Biologia Estrutural e Bioimagens e Centro Nacional de Biologia Estrutural e Bioimagens, Universidade Federal do Rio de Janeiro, Rio de Janeiro 21941-901, Brazil

**Keywords:** anaerobic parasites, pathogenicity, cell communication, cadherins, high-resolution SEM, *Trichomonas* adhesion

## Abstract

*Trichomonas vaginalis* is an extracellular protozoan parasite that causes human trichomoniasis, a sexually transmitted infection (STI) that affects approximately 270 million people worldwide. The phenomenon of *T. vaginalis* adhesion to inert substrates has been described in several reports. Still, very few studies on cluster formation have been conducted, and more detailed analyses of the contact regions between the parasites’ membranes in these aggregate formations have not been carried out. The present study aims to show that *T. vaginalis* forms a tight monolayer, similar to an epithelium, with parasites firmly adhered to the culture flask bottom by interdigitations and in the absence of host cells. In addition, we analyzed and compared the formation of the clusters, focusing on parasite aggregates that float in the culture flasks. We employed various imaging techniques, including high-resolution scanning electron microscopy, transmission electron microscopy, cytochemistry, TEM tomography, and dye injection. We analyzed whether the monolayer behaves as an epithelium, analyzing cell junctions, cell communication, and ultrastructural aspects, and concluded that monolayer formation differs from cluster formation in many aspects. The monolayers form strong adhesion, whereas the clusters have fragile attachments. We did not find fusion or the passage of molecules between neighbor-attached cells; there is no need for different strains to form filopodia, cytonemes, and extracellular vesicles during cluster and monolayer formation.

## 1. Introduction

*Trichomonas vaginalis* is a microaerophilic and extracellular parasite that colonizes the urogenital tract of humans (Figure 1, Figure 2 and Figure 3). Trichomoniasis affects more than 270 million people worldwide and is the third-most common sexually transmitted infection. As a result, women present frequent miscarriages, vaginal odor, and discharge, which can lead to infertility [1,2]. In addition, men, although usually asymptomatic, can present urethritis and prostate cancer [3,4]. *T. vaginalis* has also been found in the respiratory tract of adults and children [5,6]. Furthermore, trichomoniasis has been linked to an increased risk of HIV (human immunodeficiency virus) and HPV (human papillomavirus) [7]. Despite the health problems provoked by *T. vaginalis*, it continues to be a highly neglected organism and poorly studied. The parasite does not invade the host cells and instead attaches firmly to the mucosa to obtain nutrients and achieve successful infection. Thus, adhesion proteins play a pivotal role in the interaction with host cells, exemplified by TvAP65 and TvAP33, which inhibit host cell proliferation and can induce apoptosis. Blocking this protein has been shown to reduce its pathogenicity [8,9,10]. There is an association between cluster formation in strains with higher adhesive capacity; however, the mechanisms behind this phenomenon are relatively understudied [11]. It has been demonstrated that connections exist between the flagella of *T. vaginalis* and an extra-axonemal structure, which could potentially play a role in intercellular communication [12]. *T. vaginalis* possesses surface proteins capable of binding to host cell glycosaminoglycans, and it has been debated whether this interaction is a factor that increases the adherence of this parasite [13].

When *T. vaginalis* is grown in an axenic medium (TYM) [14], the cells are pyriform in shape and display free-swimming behavior. However, the parasite changes its morphology under specific situations, such as intense stress due to starvation or drug treatment. In addition, virulent parasites change to an ameboid shape when in contact with target cells [15]. In addition, after the parasites adhere to host cells, the trophozoites clump and transform into an ameboid shape [16]. Examination of human biopsies has revealed that *T. vaginalis* trophozoites cluster in small areas of the mucosa.

Several previous works debated the cytopathic effect of the parasite on host cells. It has been suggested that damage to epithelial cells by *T. vaginalis* occurs initially through the adhesion and clumping of the parasites [16,17,18]. Consequently, host cells die, and cell debris is phagocyted [15].

This study focused on forming *T. vaginalis* monolayers and cell clusters without host cells. Although observed by several authors, morphological data and other details were not analyzed at the ultrastructural level. In addition, as *T. vaginalis* was found firmly adhered to each other, forming interdigitations, we investigated whether the monolayer behaved as an epithelium and if communication between neighbor parasites could occur during cellular contact. It is important to note that most *T. vaginalis* adhesion and infection studies focus on the parasite’s adhesion to host cells. Here, we focus on the adhesion of trichomonas to each other, forming a united group. These aspects have received very little attention. Therefore, morphological and molecular analyses of the parasite’s mechanisms to act together, and thus be successful in adhesion and infection, contribute significantly to promising studies in the parasite’s cellular biology.

## 2. Materials and Methods

### 2.1. Parasite Culture

The JT strain of *T. vaginalis* was isolated at the Hospital Universitário, Universidade Federal do Rio de Janeiro (Rio de Janeiro, Brazil), and is a low virulent strain. The FMV1 strain is a fresh isolate kindly provided by Dr. J. Baptista (Instituto Oswaldo Cruz, Rio de Janeiro, Brazil). Dr. John F. Alderete kindly provided the fresh isolate T068. Based on its ability to destroy cells in culture, JT was previously classified as a low cytotoxic strain, whereas FMV1 and T068 were defined as cytotoxic strains [7]. The cells were cultivated in 15 mL Falcon^®^ tubes containing TYM medium [14] at pH 6.2. The medium was supplemented with 10% FBS (Thermo Fisher, Waltham, MA, USA). The cells were maintained in 37 °C incubators for 24–36 h until they reached 80–100% confluency. Sterilized 13 mm round coverslips were placed inside a 24-well plate. The cells were detached from the tube wall by placing the tube on ice for 10 min and then centrifuging it at 1000× *g* for 5 min. The cells were washed twice with phosphate-buffered saline (PBS) at pH 7.2 and 37 °C and centrifuged. The pellet was resuspended in 400 µL of the supplemented medium. Subsequently, 100 µL of the cell suspension was pipetted onto each coverslip, and the well opening was sealed with Parafilm to reduce oxygen exposure. The coverslips were incubated in a 37 °C incubator for 2–3 h and fixed as necessary.

### 2.2. Fraction of T. vaginalis Cytoskeleton

As described above, the trophozoites were detached from the tube wall and washed twice in PBS (pH 7.2) with centrifugations at 1000× *g* for 5 min at room temperature (RT). For the cytoskeleton extraction, the washed cells were then resuspended in 1 mL of ice-cold PHEM buffer (60 mM PIPES, 25 mM HEPES, 10 mM EGTA, and 2 mM MgCl_2_) containing 30% glycerol (Sigma–Aldrich, EUA), 2% Triton X-100 (Sigma–Aldrich, Saint Louis, MO, USA), 2% Igepal (Sigma–Aldrich), and 1 complete mini protease inhibitor (Sigma–Aldrich). The solution was vigorously vortexed at maximum speed for 2 min, incubated on ice for 2 min, and this was repeated two times. Then, the cytoskeleton-enriched fraction of *T. vaginalis* was washed in PBS (pH 7.2) with centrifugation at 17,000× *g* for 5 min at 4 °C. The success of the extraction was confirmed through light microscopy.

### 2.3. Cluster Formation

The tube containing the culture was placed in ice for 15 min to allow cell detachment from the tube’s walls, followed by centrifugation at 2100 rpm for 5 min. The resulting pellet was resuspended in 1 mL of TYM medium supplemented with fetal bovine serum (FBS), and cell counting was performed, yielding a result of 9.10^6^ cells/mL. Subsequently, a volume of 300 μL was inoculated into each 35 mm Petri dish (utilizing three dishes). The volume was adjusted to 4 mL and sealed with parafilm to achieve a reduced oxygen environment. The behavior of the cells was observed under an inverted light microscope. Within 5 min, all three Petri dishes already contained adhered cells on the plastic surface, exhibiting an ameboid shape with a 10–15% confluence. Plate 1 contained small clusters composed of 4–5 cells each.

Similarly, plate 2 presented small clusters, with a maximum of 10–12 protozoa per cluster. Plate 3 displayed a profile akin to that of plate 1. Subsequently, the plates were placed in an incubator at 37 °C for 10 min. After this period, the supernatant of all plates contained 80–90% clusters, while the confluence of adhered cells reached approximately 50%. Plate 1 exhibited numerous large clusters, each comprising 100 or more cells. Plate 2 displayed many medium-sized clusters (40–50 protozoa), with fewer large clusters than plate 1. Plate 3 exhibited a profile like that of plate 2. The plates were then subjected to another 15 min of incubation at 37 °C. The observed pattern closely resembled the previous analysis, with the notable distinction that the clusters present in all three plates were predominantly large, each containing more than 100 cells. Subsequently, the supernatant from the plates was carefully collected and transferred to 15 mL tubes, followed by centrifugation at 800 rpm for 5 min. The resulting pellet was fixed with 2.5% glutaraldehyde in 0.1 M sodium cacodylate buffer for 2 h at RT, without prior washing steps, to prevent cluster disassociation.

### 2.4. Ruthenium Red

Cells were fixed at RT in 2.5% glutaraldehyde in a 0.1 M cacodylate buffer containing 1% ruthenium red (British Drug Houses, Ltd., London, UK). The cells were washed with buffer and 1% OsO_4_ containing 1% ruthenium red, dehydrated in acetone, and embedded in Epon. Ultrathin sections were harvested on 300 mesh copper grids and observed without staining.

### 2.5. Thiéry Technique [19]

For TEM, the parasites were fixed and processed as described above. Subsequently, 90 nm sections were collected on gold grids incubated for 20 min in a solution containing 1% periodic acid, washed, and incubated with 1% thiosemicarbazide in 10% acetic acid for 24 h. Successive washes were carried out in 10%, 5%, and 2% acetic acid for 10 min each. Afterward, they were incubated with 1% silver proteinate for 30 min and protected from light. Subsequently, successive washes in distilled water were performed for 10 min each, and the unstained sections were observed on a Hitachi HT 7800 transmission electron microscope.

### 2.6. Immunolabeling

Cells were fixed overnight at RT in 0.5% glutaraldehyde plus 2% formaldehyde in a 0.1 M cacodylate buffer, washed, dehydrated in acetone, and embedded in LR white resin. Ultrathin sections were harvested on 300 mesh, and labeling proceeded using the antibodies cited in the immunofluorescence section above.

### 2.7. Electron Microscopy Tomography

As described above, cells were processed for TEM, and blocks were used to obtain 200 nm thick sections, which were collected onto formvar-coated copper grids and stained with uranyl acetate and lead citrate. For alignment of the tilted views, colloidal gold particles (10 nm) were deposited onto both sections’ surfaces to be used as fiducial markers. A single-axis tilt series (±)55° with 2° increments was produced from samples using Xplore D software and a Tecnai Spirit TEM (Thermo Fisher, Waltham, MA, USA) electron microscope operating at 120 kV. The IMOD software package performed all 3D reconstructions and subsequent 3D data analyses. ETOMO was used to generate a tomogram by R-weighted back projection. Virtual slices were manually segmented with 3DMOD to produce 3D models.

### 2.8. Dye Injection

Confluent cultures of *Trichomonas vaginalis* plated on 35 mm Petri dishes were injected with Lucifer Yellow CH (5% in 150 mM LiCl) (457.2 Da) using glass microelectrodes (resistance between 40 and 70 MΩ) by short hyperpolarizing current pulses (0.1 nA, 100 milliseconds using a WPI amplifier, model 7060; USA). Fluorescence was observed on an Axiovert 100 microscope (Carl Zeiss, Oberkochen, Germany) equipped with appropriate filters (Zeiss BP450-490/FT510/LP520), and micrographs were taken using the Image Pro Plus program (Media Cybernetics, Rockville, MD, USA) 2 min after dye injection [20]. A minimum of 120 cells were injected in at least four independent experiments to determine the degree of coupling.

## 3. Results

### 3.1. The First Cell Contacts

In the present study, we intended to follow the formation of the trichomonads monolayers and clumps using light and electron microscopy, among other techniques. Concerning the formation of the monolayer, we observed that the parasites were well spread out on the inert surface of the bottom of the flasks used. First, parasites projected cell surface expansions and firmly adhered to the inert substrate. A transition from a pear-shaped to an amoeboid form was observed, and the cells spread with such intensity that they became flattened (Figure 1). The cell contacts were examined by high-resolution scanning and transmission electron microscopy, revealing the formation of plasma membrane contact and interdigitations. Initially, the single cells exhibited a routine morphology, presenting four anterior flagella and one recurrent flagellum, and were pear-shaped (Figure 1).

We observed a gradual contact between single cells with the bottom flask in all experimental assays. Within a few seconds, the parasite emitted cell surface projections, such as filopodia, lamellipodia, and pseudopods (Figure 1 and Figure 2). The cells were so firmly adhered to the bottom flask that even several washes did not remove them. One interesting observation was that the undulating membrane was expanded and did not participate in the contact with the neighboring cells (Figure 2b). The flagella touched a neighboring cell and, thus, seemed to participate in cell recognition and the subsequent approach of the two cells (Figure 2e).

More cells gradually approached each other, adhered, and formed a monolayer (Figure 2, Figure 3 and Figure 4). It is important to mention that a second layer of parasites adhered when the monolayer was formed (Figure 4). We tested different strains with low and high virulence. When in contact with the flask bottom, all strains adhered. The analyses were conducted using light, scanning (SEM), and transmission electron microscopy (TEM) (Figure 1, Figure 2, Figure 3 and Figure 4). The adhesion between the cells was strong, and the cells did not separate when the monolayer was scraped from the flask. It was not necessary to add host cells to induce monolayer formation. Observing the samples with SEM revealed that the parasites were initially pear-shaped (Figure 1b–e and Figure 2). However, they become gradually ameboid, flattened, and spread over the inert surface (Figure 2, Figure 3 and Figure 4). During monolayer formation, all flagella were kept upward and never in contact with the flask bottom (Figure 1 and Figure 2).

### 3.2. Monolayer Formation and Cell–Cell Attachment

TEM analyses revealed the first observations of plasma membrane contacts, followed by several close contacts between the parasites (Figure 2, Figure 3, Figure 4 and Figure 5). The membranes were very close in several images and appeared fused in some situations. Then, we used markers such as ruthenium red (Figure 3d) and the Thiéry technique (Figure 3e–f) to check if there was membrane fusion and if there would be passage of molecules between the cell–cell contact areas. Furthermore, we obtained tomograms in regions where there was doubt about whether fusion would occur (Figure 5). Thus, the interaction areas were aligned using thick slices. With a 120 KV TEM acceleration, sequential and deep images were obtained. What appeared to be a fusion between cell membranes turned out to be cells with their membranes intact, without fusion occurring, at least in the cells tested (Appendix A). Thus, no fusion of the parasites’ cell membranes forming the monolayer appeared to occur. However, new studies are needed to confirm these findings.

Electron tomography enabled the visualization of an inward folding or invagination in the lower layers. Upon observing the tomographic reconstruction video, it became evident that reconstructing the image was crucial to accurately perceive the folding of the plasma membrane, thereby preventing any confusion with a potential fusion of regions (Appendix A).

### 3.3. Extrusion of T. vaginalis Extracellular Vesicles (TvEVs)

During the formation of the monolayer, we observed multivesicular bodies in the cells’ internal vacuoles (Figure 3). In addition, extracellular vesicles in the extracellular environment, in close contact with other parasites, were also observed in large amounts (Figure 4). Interestingly, there were no host cells, only other parasites of the same strain.

### 3.4. Proteins of Junctional Areas

We also decided to look for known proteins present in the junctional areas of the cells that form epithelia in metazoan cells. We tested for claudin, occludin, and ZO-1 with commercial antibodies—all results were negative. Regarding tests with anti-cadherin antibodies, we started with a recent group publication [21], which found a protein similar to cadherin using bioinformatics. We used various commercial pan-cadherin antibodies. A positive reaction was observed, albeit not in the cell–cell contact regions, but in the anterior portion near the communication interface between the cytoskeleton and the plasma membrane (Figure 6 and Figure 7, Appendix A). When we tested enriched fractions of the *T. vaginalis* cytoskeleton after detergent-based extraction using anti-tubulin and anti-cadherin antibodies, we noticed a co-localization of the labeling (Figure 7), indicating a non-specificity of the commercial anti-cadherin antibodies.

Labeling observed in whole cells and isolated cytoskeletons of *T. vaginalis* with the E-cadherin polyclonal antibody (Invitrogen, Waltham, MA, USA) may not target a membrane or cadherin-like protein. Additionally, we conducted membrane labeling using Concanavalin A, which recognizes alpha-mannose residues (Figure 6), and the labeling pattern remained consistent, suggesting an association with the cytoskeleton rather than the membrane or the cell–cell contact region.

### 3.5. Communication between Parasites: Injection Tests

In order to verify whether there was a passage of information between adjacent cells that formed the monolayer, as occurs in gap junctions, we performed an experimental injection of the fluorescent dye Lucifer Yellow in one cell and took pictures 2 min later to see if the dye was found in neighboring cells—the results were negative. The dye was restricted to the injection site (Figure 8).

## 4. Cluster Formation

While cultivating different strains of T. vaginalis, we observed the formation of cell aggregates that can reach hundreds of attached parasites (Figure 9). It was observed that contacts occur without the formation of interdigitations, as in the formation of monolayers. Furthermore, the aggregates were fragile and dispersed after vigorously shaking the tubes, which did not occur with monolayers when the parasites adhered to the inert material of the tube walls.

## 5. Discussion

*T. vaginalis* is an extracellular parasite that uses adherence as a crucial factor for its pathogenicity. Many studies have reported molecules and factors indispensable for infection [22,23]. Those that analyze the behavior and mode of parasites adhering to one another are very important and help elucidate parasite survival and infection.

In the present work, we analyzed the formation of monolayers and parasite clusters using high-resolution scanning electron microscopy, among other techniques. Even though it was an inert material, the parasites came into contact with membranes and formed a uniform layer of tightly adherent cells, similar to an epithelium. Based on this observation, we proceeded to analyze whether (1) there would be a fusion between the cell membranes of the parasites, (2) there would be a passage of molecules between the contact areas, and (3) there would be a passage of molecules between the cells of the monolayer, as occurs in gap junctions. We used advanced microscopic analysis, such as high-resolution SEM and TEM immunostaining using antibodies against junctional area proteins found in epithelial cells, ultrastructural tomography, and fluorescent dye injection to check if parasite communication occurred.

### 5.1. Monolayer Formation by T. vaginalis

In the present work, we observed that when parasites formed monolayers, extended areas in neighboring cells were tightly attached by multiple interdigitations of the adjacent cells. The flat ridges of neighboring cells were intensely interlocked, and the intercellular spaces were closed.

In metazoa, cell junctions of different types are responsible for several functions, such as mechanical, chemical, and electrical coupling of cells and forming particular barriers in epithelia and endothelia. In the extended areas of the small intestinal epithelium, neighboring cells are tightly attached, and the epithelium is stabilized by multiple interdigitations of the adjacent cells. In kidney tubules, these interdigitations are characterized by an extracellular space of relatively constant width (about 50 nm), separating the parallel membranes of adjacent cells.

Although single-celled eukaryotes, such as *T. vaginalis*, do not contain cell junctions, we demonstrated the formation of cell–cell junctional regions represented by interdigitations, similar to metazoa epithelia in parasites when in contact with an inert substrate. Previous work has also reported the presence of interdigitations between parasites when in contact with host cells [16].

Here, we show that a protist, *T. vaginalis,* can establish a monolayer similar to metazoa epithelia, with tight adhesion between cells via interdigitations, playing a significant role in parasite attachment to and possibly aiding the lysis of host cells. Because several cells can press the host cell with greater intensity, the monolayer and clusters of parasites can provide great resistance and stability, favoring the parasites as they can cause greater damage to the vaginal epithelium. In addition, the increased parasite–parasite association may further increase the number of parasites attacking the host cells. 

It has been demonstrated in several works that *Trichomonas vaginalis* can adhere to plastic surfaces in the presence of various agents and under different experimental conditions [24,25,26,27,28]. In addition, one group [28] demonstrated that the parasite can adhere to polystyrene, intrauterine devices, and vaginal rings. However, although another group [26] reported that actin was important in the adhesion of neighboring cells, a detailed ultrastructural report has not been published. When the authors compared *T. vaginalis* adhesion to an inert substrate with the host cell, they discovered that high adhesin gene expression levels were observed only in trophozoites attached to the cells [29]. One study [27] compared the in vitro adhesion of *T. vaginalis* to plastic, testing parent and lipophosphoglycan (LPG) mutants, and discovered that they differed in their attachment to plastic surfaces.

Electron tomography microscopy has thoroughly delineated the intricate three-dimensional structures of diverse cellular models, including parasites. These methods encompass 3D reconstruction from consecutive sections, diverse electron tomography techniques such as STEM and serial tomography, and, more recently, scanning electron microscopy combined with a focused ion beam (FIB) [30,31,32]. Nevertheless, few studies incorporating tomographic analyses have focused on examining *Trichomonas vaginalis*, indicating a promising avenue for applying this technique.

Electron microscopy has revealed the intricate structural details of this parasite, facilitating investigations into both its two-dimensional (2D) and, less frequently, three-dimensional (3D) ultrastructural attributes [33]. This comprehensive approach has contributed immensely to our understanding of *Trichomonas* biology and pathogenesis, shedding light on various aspects crucial for research and clinical perspectives. Nevertheless, despite the substantial progress and technological advancements, particularly in electron tomography (ET), there remains a notable absence of tomographic analysis in most studies focused on ultrastructural examinations of *Trichomonas*. This absence points to a significant gap in our comprehension of the parasite’s complete three-dimensional structure and organization. This underutilization of tomography analysis restricts the comprehensive exploration of the parasite architecture, impeding a deeper understanding of its cellular intricacies, spatial relationships, and potential dynamic interactions within its microenvironment. The paucity of studies embracing tomographic methods overlooks the opportunity to employ cutting-edge techniques that could enhance our understanding of *Trichomonas* ultrastructure. The 3D perspectives derived from tomographic analyses could offer an unparalleled advantage in deciphering complex cellular arrangements, internal structural details, and the spatial distribution of various components within *Trichomonas* that might not be fully appreciated in conventional 2D assessments. Thus, the limited application of tomographic analysis in studies of *Trichomonas* represents an untapped potential in comprehensively unraveling its intricate three-dimensional architecture. Incorporating advanced tomographic methods could significantly advance our understanding of the parasite ultrastructure and cellular dynamics, potentially opening new avenues for targeted research and therapeutic interventions in *Trichomonas*-associated diseases.

### 5.2. Clumping

The observation of clumps has already been reported in previous articles, suggesting that aggregation may have a role in the pathology of *Trichomonas* [16,18,21]. However, the importance of clumping and the stages of clump formation and cell-to-cell union under morphological aspects have not yet been described. Previous works defined a clump as an aggregate of ∼ 10 or more parasites [18,21]. Here, using electron microscopy, it was possible to see that hundreds of parasites can form a clump. We observed that the first clumps are easily disrupted by pipetting, indicating a fragile adhesion, which differed from the observation on monolayers, which are firmly adhered.

One group [18] claimed that TvTSP8 from the Tetraspanin family (TvTSPs) was involved in parasite aggregation, suggesting a role for this protein in parasite interaction. In addition, it has been reported that cytonemes are associated with clump formation [34] and that highly adherent strains aggregate more than poorly adherent strains [18,34,35]. The authors demonstrated that different parasites inside the clumps are connected by cytonemes, which are longer filopodia extensions reaching up to ~300 nm from the originating cell body [34]. The authors reported that interaction between different strains induced cytoneme formation [36] and that parasite clumping is strain-dependent. In addition, the size-variable microcolonies dysregulate epithelium permeability or promote its destruction. However, the authors did not show references for these disease facts, although it was not a full paper but a monthly commentary on parasites. In addition, the authors reported that different *T. vaginalis* strains communicated through cytoneme-like membranous cell connections, and that the cytoneme formation of an adherent parasite strain is affected in the presence of a different strain [34]. In our work, filopodia and cytonemes are longer than 6 µm, and we showed no need for different strains to form filopodia, cytonemes, and extracellular vesicles.

Several works have reported the transition from pear-shaped parasite to amoeboid, including the actin-based machinery and parasite migration across host tissue [26,37]. One group [16] reported that axenic trophozoites clumped within the first 15 min of contact after adhering to target cells and transformed into an ameboid shape. However, the authors did not focus on monolayer formation and the parasite’s adhesion to inert substrates.

Defining the molecules and the morphological characteristics that *T. vaginalis* uses may help us understand how the parasite colonizes the urogenital tract and how to prevent or treat infections.

Some authors have described the importance of the virulence of the strain [18,21,36] as a determining factor for the formation of clumps, indicating that calcium is also an important ion for that. In the present work, we did not verify these indications. All strains analyzed showed clumps, regardless of whether they had high or low virulence. However, the strains we used were different from the other groups.

The Pachano group [17] reported that increased histone acetylation leads to increased parasite aggregation and adherence to host cells. The authors claimed that TSA treatment is responsible, at least partly, for increased parasite clumping and adhesion of the parasite to human host cells. In addition, Coceres et al., 2015 [18] showed that increased TSP8 expression increased parasite aggregation in TSA-treated parasites.

Recently, epigenetic regulation for *T. vaginalis* cytoadherence and parasite aggregation has been proposed [28]. The authors reported that when a less-adherent strain was treated with the histone deacetylase inhibitor trichostatin A (TSA), cytoadherence and parasite aggregation were improved, lending credence to the significance of their findings. In addition, galectins binding to *T. vaginalis* surface lipoglycans have been reported as one of the potential factors that triggers parasite swarming [38]. Thus, one group [28] applied lactose to compete with the interaction between galectin and the parasite ligand. The authors observed that the parasite clusters on host cells were significantly reduced, indicating that lactose could compete with parasite aggregation.

Although it has been reported that clumping does not occur in all parasite strains, we discovered that cell-to-cell adhesion to form an adhesive parasite monolayer occurred in all of the strains we studied. As far as we know, this is the first study showing that *T. vaginalis* forms a very tight monolayer before clumping when in contact with an inert substrate.

The association of clumping with parasite adherence is interesting in light of emerging evidence suggesting that aggregation may have a role in the pathology of *Trichomonas*. Defining the biochemical properties required for adhesive phenotypes of *T. vaginalis* may help us understand how the parasite colonizes the urogenital tract and how to prevent or treat infections.

### 5.3. E-Cadherin

A previous bioinformatic work [21] identified a hypothetical protein, TVAG_393390, which was renamed cadherin-like protein (CLP). Its predicted tertiary structure was similar to mammalian cadherin proteins involved in cell–cell adherence. The authors showed that *T. vaginalis* overexpressing CLP has a ∼3.5-fold greater adherence to host cells. These analyses described the first parasitic CLP.

In addition, one group [21] demonstrated that adding Ca^2^ had a notable impact on the clumping of parasites, while the presence of host cells did not exhibit a significant effect. These data together suggest that parasite clumping is calcium-dependent. In our work, we observed that calcium did not interfere in monolayer and clump formation. The fact that we used inert flasks indicated that the ability to form clumps and monolayers does not necessitate a host signal.

Because cadherin proteins are conserved metazoan proteins with important participation in cell–cell adhesion, and the Chen group [21] claimed that the cadherin-like protein is more abundant on the surface of parasites and mediates parasite–parasite and parasite–host adherence, we decided to analyze the presence of cadherins on the region of cell–cell contact in the trichomonads monolayers observed in the present study.

Our experiments were conducted using immunolabeling commercial pan-cadherin antibodies and analyzed with immunofluorescence and transmission electron microscopy. We used anti-cadherin antibodies from different sources, and the essays were repeated several times. We only found a positive reaction in the anterior portion near the interface between the cytoskeleton and the plasma membrane using the anti-E-cadherin polyclonal antibody (Invitrogen, Waltham, MA, USA, PA5-32178). Still, no labeling was observed on the parasite cell surface or in the cell–cell contact region.

We investigated the potential proteins that were labeled with this antibody. We employed the TrichDB BLAST tool to compare the peptide sequence of the commercial E-cadherin polyclonal antibody with the *T. vaginalis* genome. We identified TVAGG3_0955330 as the best match, scoring 31.6 and an E-value of 0.62. It was described as an ‘ankyrin repeat protein family,’ and computationally inferred from orthology, its predicted function is the transfer of phosphorus-containing groups. Ankyrins mediate the attachment of integral membrane proteins to the spectrin-actin-based membrane cytoskeleton [39,40]. We utilized a monoclonal anti-alpha-tubulin antibody (TAT-1, generously provided by Dr. Keith Gull) to identify the cytoskeleton and cross-checked its localization with E-cadherin polyclonal antibody labeling. Tubulin is recognized for its potential to interact with ankyrin, and previous studies have indicated its role in facilitating connections between cytoskeletal components and the plasma membrane [41,42].

In alignment with in silico analyses, we used the DeepTMHMM, a tool offered by the Technical University of Denmark employing deep learning models for transmembrane topology prediction and classification, and we determined that TVAGG3_0955330 lacks transmembrane regions [43]. Furthermore, it does not exhibit potential GPI modification sites, as suggested by the findings from the GPI Lipid Anchor Project (Institute of Molecular Pathology, Wien, Austria) [44]. These findings support the conclusion that this protein is localized within the cytosol and not integrated into the membrane. 

It is important to point out that the work of the Chen group [21] only found structural similarity with cadherins and that it is located in the cell membrane only when CLP is overexpressed. Consequently, it is difficult to assert its involvement in intercellular communication, as our understanding leans towards a more structural role, wherein this protein localizes within cells without genetic manipulation.

### 5.4. Extracellular Vesicles

Extracellular vesicles (EVs) are well-recognized mediators of intercellular communication, and the presence of vesicular bodies has attracted much research interest, mainly in studies of interactions between parasites and hosts [45]. It has been stated in published works that *T. vaginalis* releases EVs, which seem to play a role among parasites and parasite–host interactions [35,46]. The last two groups [35,46] stated that *T. vaginalis* produced and secreted microvesicles and reported the identification of microvesicle-like structures (MVs) released by *T. vaginalis*. In this study, the authors indicated that 1 mM CaCl_2_ or 1 mM CaCl_2_ + 4.5 μM calcium ionophore could induce the formation of microvesicles in the trophozoites of *T. vaginalis.* In our observations, the parasites produced microvesicles without adding CaCl_2_ or calcium ionophore. We speculate that the parasite contact with themselves favors the production of MV and triggers the extrusion, as the parasites were being prepared for the attack on host cells.

The production of vesicles of different sizes has been demonstrated when *T. vaginalis* comes into contact with the host cell and when different strains are in contact, which has indicated changes in parasite adherence mediated by extracellular vesicles [23,34,35,46,47,48]. In addition, *T. vaginalis* exosomes modulate host immune responses since they induce an IL6 response in vaginal epithelial cells and downregulate the IL8 response to parasites [46]. Thus, microvesicles and other larger vesicles produced by *T. vaginalis* provoke specific physiological stimuli [35]. *T. vaginalis* EVs can alter host–cell interactions and induce host immune responses [46,47].

*T. vaginalis* EVs have a diverse genetic material composition, primarily enriched with small RNAs [45]. It has been suggested that they could modulate gene expression in recipient cells, potentially impacting parasite–parasite or parasite–host interactions. The delivery of these small RNAs to host cells for gene activity modulation is a promising area for further research [46,49].

The Molgora group [50] recently reported that extracellular vesicles can impact *T. vaginalis’* survival. The authors demonstrated that extracellular vesicles (EVs) secreted from Tv are internalized by host cells [45,48] and increase the adherence of this extracellular parasite to host cells [46] in vitro, indicating that EVs likely assist in parasite colonization of the host in vivo and in vitro. Recently, the authors used an in vivo model for pathogenesis analyses [50]. They showed that co-inoculation of TvEVs and parasites results in higher parasite burden in vivo; thus, the presence of EVs significantly increases parasite survival in vivo, confirming a previous publication where the same group used in vitro-based predictions that TvEVs assist the parasite in colonizing the host.

The presence of EVs and their role in *T. vaginalis* when clumping during host interaction have not been deeply analyzed, and their participation in the infection process is still scarce. In the present study, we observed the presence of large intracellular vacuoles containing multivesicular bodies and also the presence of extracellular vesicles. The present study had no interaction with host cells, any other cells, or tested different strains. We observed an intense production of vesicles in sites of strong adhesion between the parasites. We suggest that microvesicles may also play a role in modulating information between parasites of the same strain, possibly exchanging important information about adherence between parasites. Thus, this is an important aspect that previous works have not investigated. The parasites exchange pieces of information and appear to exchange information among themselves, even in the absence of other strains or host cells. This could occur by eliminating microvesicles after contact between the flagella or the cell membranes of two or more parasites. In our assays, we observed that any inert material entering the culture tube produces a strong reaction from trichomonas; for example, when a needle is introduced into the culture tube, the parasites immediately adhere to it. This leads us to suggest that the parasites seem to be programmed to be edacious for any material that serves as support and contact, thus being able to trigger their mechanism of adhesion or attack on homologous, foreign cells, or even inert materials.

## 6. Conclusions

In conclusion, we show that single cells, such as the parasite *T. vaginalis,* form a tight monolayer that, at first glance, is similar to a metazoa epithelium. We compared it with *T. vaginalis* clumping and showed that this parasite behaves differently when clumping and adhering to a substrate. The trichomonas clusters float in the cell cultures, exhibit distinct plasma membrane contacts with no interdigitations as seen in monolayer formations, do not change to an ameboid shape, and have fragile connections. In contrast, the monolayers are tightly adhered and do not separate even with high-speed centrifugation. We also report that in the monolayer and cluster formation, no fusion or passage of molecules occurs in both events, although new studies are necessary to confirm these findings. In addition, our work shows no need for different strains to form filopodia, cytonemes, and extracellular vesicles during cluster formation. Extracellular vesicles were noted inside vacuoles and in the extracellular space in cells without contacting host cells. Taken together, *T. vaginalis* seems to cooperate with other parasites, which could reinforce their strength in attacking host cells.

## Figures and Tables

**Figure 1 pathogens-12-01381-f001:**
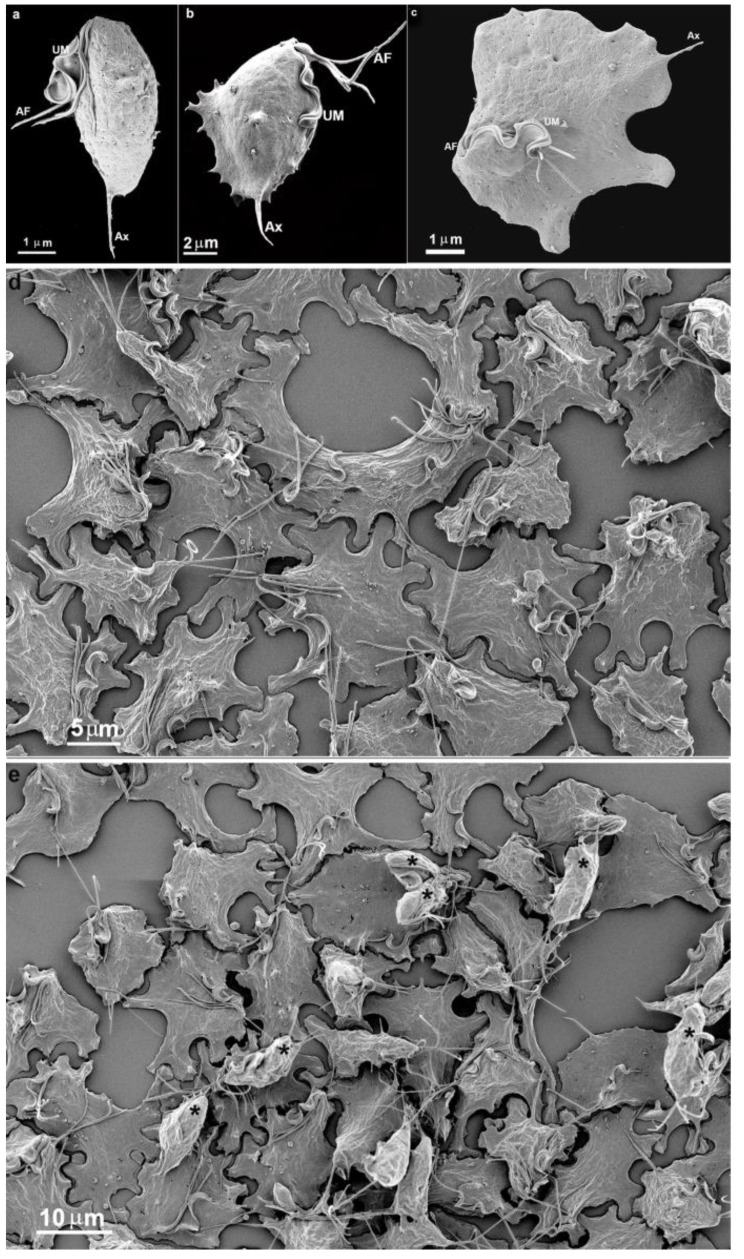
Scanning electron microscopy of the parasite *T. vaginalis* grown in axenic culture (**a**) and after contact with an inert substrate or host cell (**b**,**c**). (**b**) A cell was taken almost immediately after adhesion to the flask bottom. Notice that small cell-surface projections are seen, which do not occur in free cells. (**c**) An ameboid *T. vaginalis*. (**d**,**e**) SEM of *T. vaginalis* as a monolayer. Notice the flattened parasites presenting cell contacts with neighbors. In (**e**), a second layer of parasites was caught in the adhesion process (asterisks). AF: anterior flagella; Ax: axostyle; UM: undulating membrane.

**Figure 2 pathogens-12-01381-f002:**
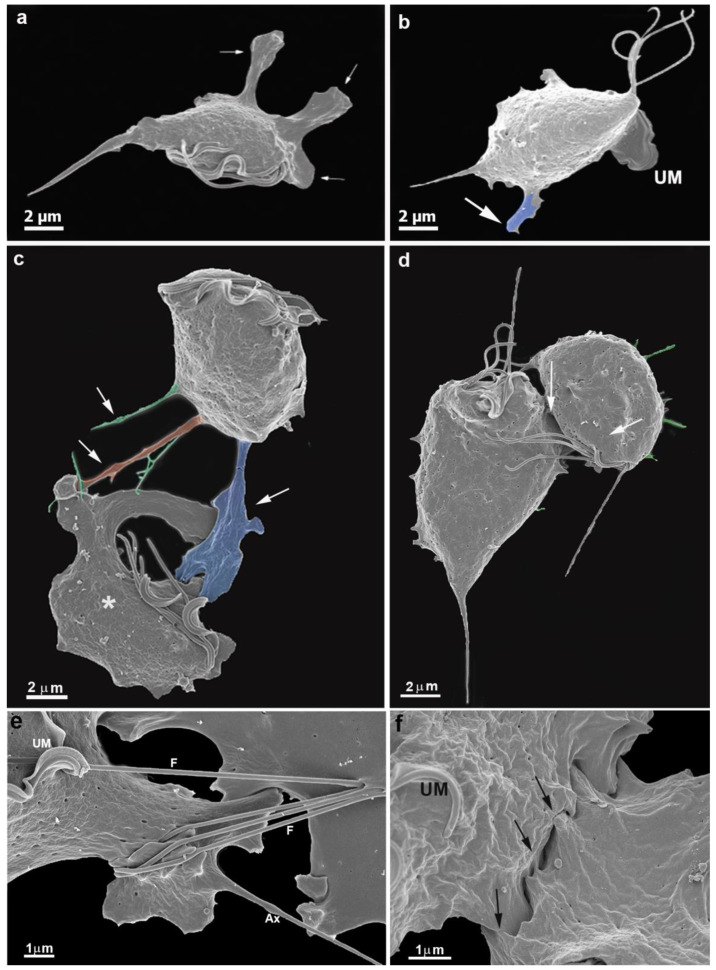
Scanning electron microscopy of *T. vaginalis* collected from the flask bottom after a few minutes of cultivation. Parasites were fixed in situ, washed several times, and processed for SEM. Large cell projections in blue (**a**–**c**), filopodia (orange), and cytonemes (green). (**c**,**d**) Early contacts between two parasites (arrows). Notice cell projections and changes in the shape of one cell (**c**, asterisk). (**e**,**f**) Cells in contact. Notice the flagella contacts with a neighboring cell (**e**) and close contacts between cell surfaces (**f**). UM, undulating membrane.

**Figure 3 pathogens-12-01381-f003:**
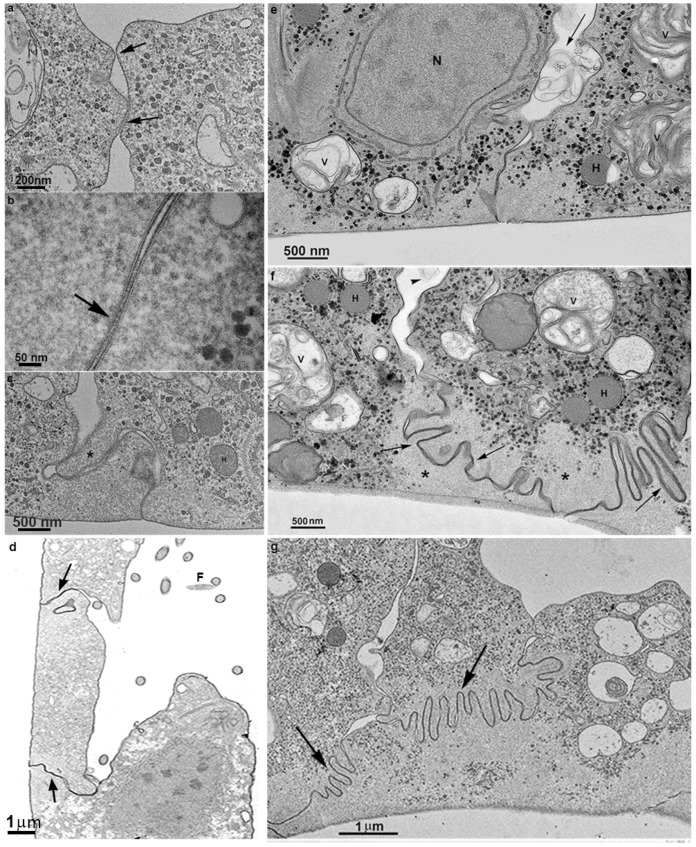
TEM of sequential images of the first *T. vaginalis* cell contacts until the formation of the monolayer on the flask bottom. (**a**,**b**) Plasma membranes are in contact. Notice the glycocalyx contacts. (**c**) Beginning of the formation of the first interdigitations (asterisk). (**c**) New and deep interdigitations form, promoting a strong bond between cells (arrows). (**d**) *T. vaginalis* incubated with ruthenium red and not stained with uranyl acetate. The arrows point to the contact between the plasma membranes, indicating that the stain could pass between the cells. (**e**,**f**) TEM of parasites in contact forming a monolayer after utilizing the Thiéry technique, which reveals carbohydrates. Notice the positive reactions in glycogen granules (black dots) and cell membranes. Many vesicles (V) containing multivesicular bodies are also observed in the extracellular region (arrows). (**g**) The cells are already tightly bound together, forming a monolayer. Note that just below the cell surface is an area devoid of organelles (asterisks in (**f**)). H: hydrogenosome; F: flagella; N: nucleus.

**Figure 4 pathogens-12-01381-f004:**
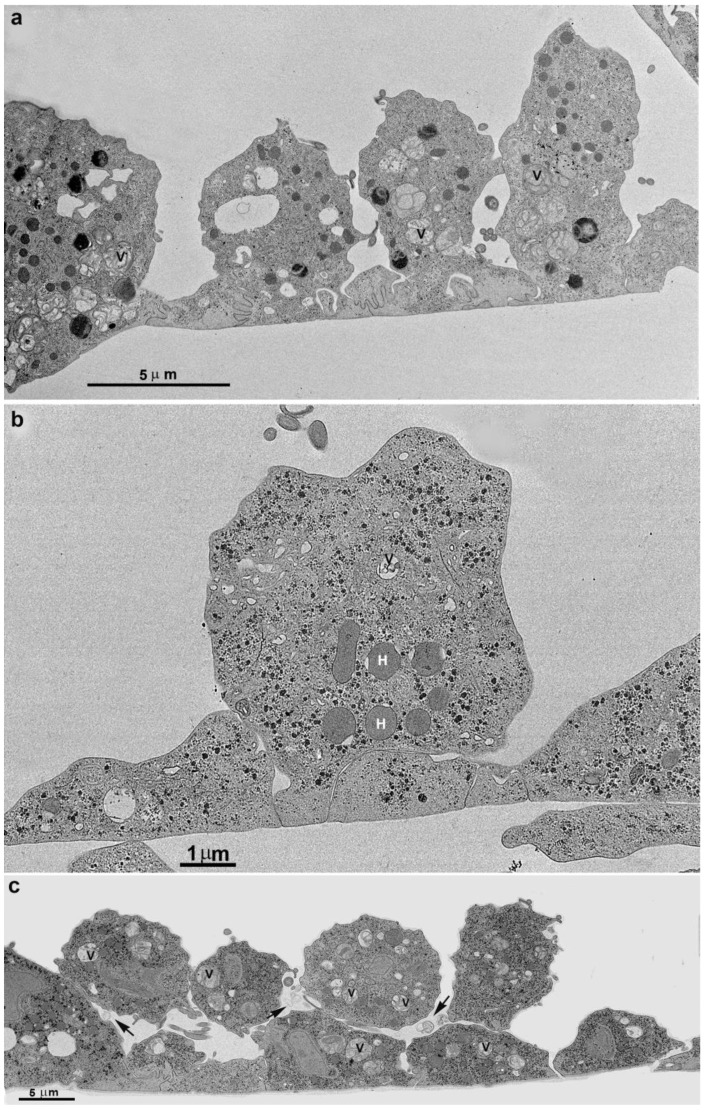
A monolayer of adhered parasites after incubation in a plastic flask. (**a**) The cells are firmly adhered to, forming many interdigitations. (**b**) In this stage, only one layer is seen. Afterward, in (**c**) a second layer is formed, and new cells are adhered to the first layer. Notice the presence of multivesicular bodies in vesicles inside the cells and the extracellular region (arrows). H, hydrogenosomes.

**Figure 5 pathogens-12-01381-f005:**
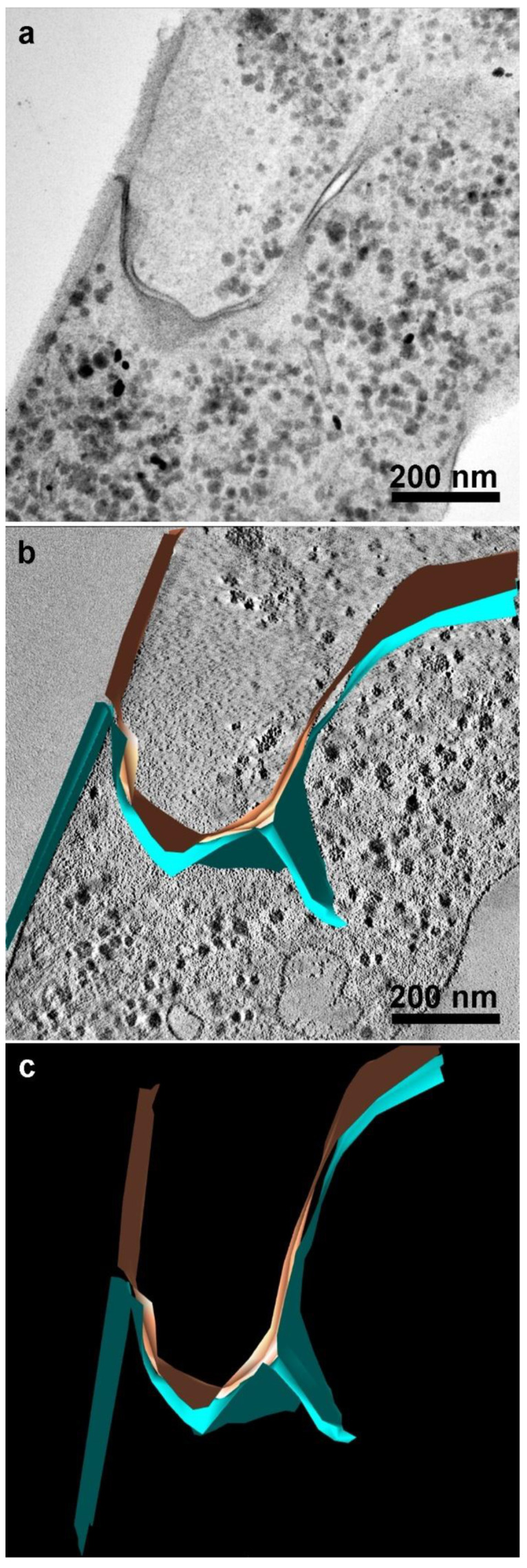
(**a**) Transmission electron microscopy of *T. vaginalis* in a region of two cells in contact in monolayer formation. (**b**) Virtual slices from a tomogram were obtained by TEM tomography, where a closer region was reconstructed and colorized. (**c**) Three-dimensional model of the tomogram. The two plasma membranes are shown in brown and blue; no membrane fusion was noticed.

**Figure 6 pathogens-12-01381-f006:**
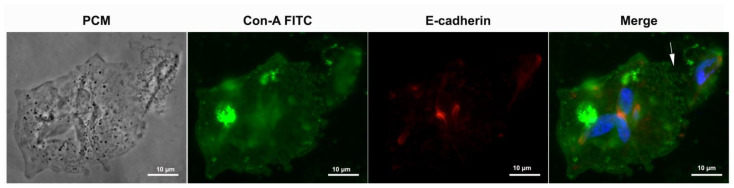
*T. vaginalis* in contact was seen by phase contrast microscopy (PCM) and incubated with fluorescent Con-A (labeling alpha-mannose residues in green) and E-cadherin polyclonal antibody (red). Notice that there is no co-localization of E-cadherin with the plasma membrane. The white arrow indicates the interdigitations between two cells. The nuclei are stained with DAPI (blue).

**Figure 7 pathogens-12-01381-f007:**
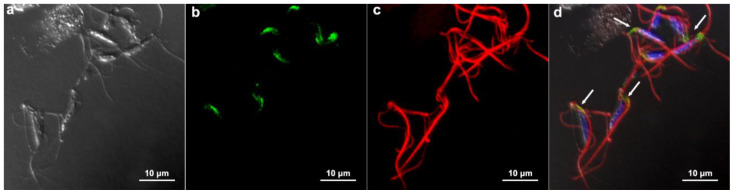
Enriched fraction of the *T. vaginalis* cytoskeleton after detergent-based extraction. (**a**) Differential interference contrast microscopy; (**b**) immunofluorescence microscopy using an anti-E-cadherin polyclonal antibody (green); (**c**) labeling with an anti-alpha-tubulin (red). In (**d**), note the co-localization of the labeling (white arrows). The nuclei remnants are stained with DAPI (blue).

**Figure 8 pathogens-12-01381-f008:**
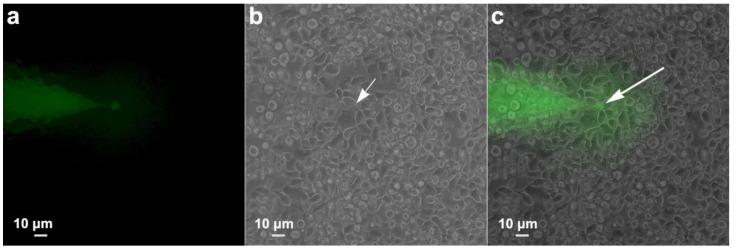
Functional analysis of a possible gap-junction coupling among *T. vaginalis* monolayer. Lucifer Yellow (green) was injected in one cell, and pictures were taken 2 min later. (**a**) Fluorescence imaging, (**b**) phase contrast, and (**c**) merged. Notice that the dye remained restricted to the injected cell (arrow).

**Figure 9 pathogens-12-01381-f009:**
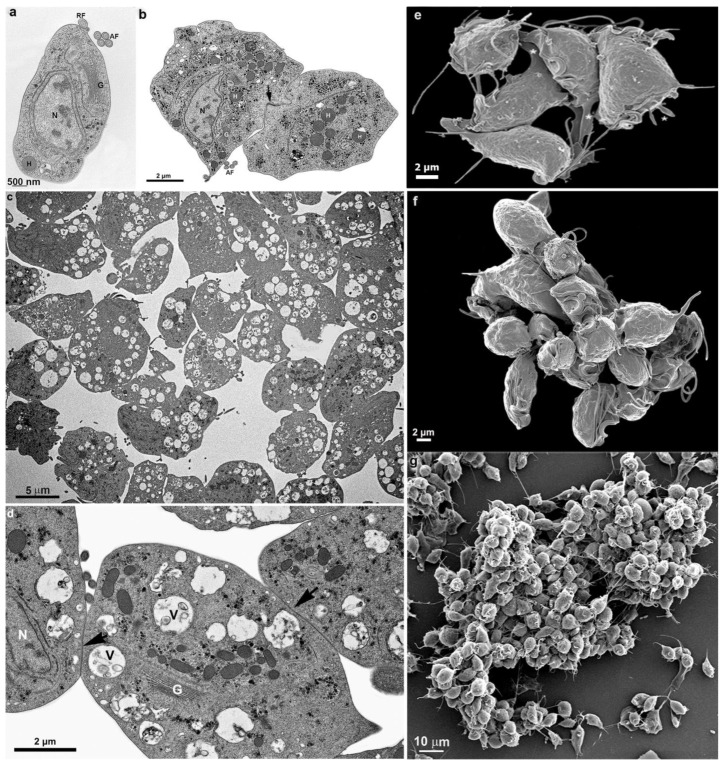
Transmission (**a**–**d**) and scanning electron microscopy (**e**–**g**) of *T. vaginalis* from the supernatant culture. (**a**) Before contact with other cells, (**b**) early cell contact, and (**c**,**d**) after a cluster formation. The arrows point to a region of adhesion. (**e**–**g**) SEM shows large clusters formed by hundreds of *T. vaginalis* (**g**). AF: anterior flagella; G: Golgi; H: hydrogenosome; N: nucleus; RF: recurrent flagellum; V: vacuole with multivesicular bodies. Asterisks indicate plasma membrane projections.

## Data Availability

The data presented in this study are available on request from the corresponding author.

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
