# Peer review of "Trichomonas vaginalis: Monolayer and Cluster Formation—Ultrastructural Aspects Using High-Resolution Scanning Electron Microscopy"

_pathogens, 2023, doi:10.3390/pathogens12121381_

Round 1

Reviewer 1 Report

Comments and Suggestions for Authors

The article entitled T.vaginalis: Monolayer and cluster formation-ultrastructural aspects using HR-SEM is very interesting and important for the field of parasitology. Some comments should be considered: 

The title should have the complete name of the parasite.

The conclusion section is repeated in the manuscript

Author Response

Reviewer 1

"The title should have the complete name of the parasite. The conclusion section is repeated in the manuscript."

Answer: Thank you for your feedback. We changed the title, added the full scientific name, and corrected the conclusions section.

Reviewer 2 Report

Comments and Suggestions for Authors

Trichomonas vaginalis, as an extracellular parasite, is widely parasitic in the human urogenital tract, causing trichomoniasis. T. vaginalis infection not only causes inflammation of the urinary and reproductive tract, but also contributes to reproductive system cancer and infertility. Understanding the morphological changes of T. vaginalis helps to understand its pathogenicity. There is currently relatively little research on the morphological changes of T. vaginalis. This manuscript is very meaningful in studying the morphological changes of T. vaginalis through techniques such as scanning electron microscopy and projection electron microscopy. However, there are many confusing aspects in this article.

Major issues

1. The structure of this manuscript does not meet the requirements of a research-oriented article. Please refer to the published research articles for modification.

2. The author has repeatedly demonstrated T. vaginalis morphology of monolayer and cluster through electron microscopy. However, this is only a phenomenon and does not solve a scientific problem.

3. In the results section, the author did not fully describe the results of each section, only briefly listed them.

4. The author should carefully consider what kind of scientific problem or phenomenon they want to illustrate through these experiments. The entire manuscript lacks logic, as if the author randomly listed some experimental results.

5. In fact, this manuscript only describes the morphology of T. vaginalis in different states. It is difficult to determine whether there is mutual communication between T. vaginalis through these electron microscopy photos. Figure12 and 13 can only illustrate the localization of E-cadherin protein in trophoblasts, as fluorescence staining of the cell membrane and cytoskeleton was not performed, making it impossible to determine whether E-cadherin is at the junction between cells. The experiment in Figure 14 lacks rigor, and its negative or positive results cannot determine whether there is communication between trophozoites. These fluorescent dyes may flow out of the trophozoite and exist in the culture medium, which is then randomly ingested by other trophozoites.

6. There are too many images in this manuscript. It is recommended to merge them into less than 5 images.

7. Suggest changing this manuscript into a short communication for publication.

8. The language of this manuscript needs to be improved.

Minor problem

1. At line 143, “9.109 cells/ml” should be modified to “9x109 cells/ml”.

2. At line 164, inconsistent spacing between lines.

3. At line 166, font inconsistency.

4. At line 189, “(+ -)55° with 2° increments)” ? Half bracket, centigrade˚C.

5. “interdigitations” or “inter digitations”.

6. At line 166, text format.

7. There are two conclusions in the manuscript.

Comments on the Quality of English Language

English very difficult to understand/incomprehensible

Author Response

Reviewer 2

Major issues

  1. The structure of this manuscript does not meet the requirements of a research-oriented article. Please refer to the published research articles for modification.

Answer: We did not understand this comment. This manuscript was an invited paper to a special issue entitled "Trichomonas vaginalis infection and belongs to the Section Parasitic Pathogens. We used the journal template and followed the instructions we received. We checked again.

  1. The author has repeatedly demonstrated T. vaginalis morphology of monolayer and cluster through electron microscopy. However, this is only a phenomenon and does not solve a scientific problem.

Answer: Our field is morphology. Morphological analyses are important, and hard work in the microscopes identifies parasite details and helps us and other researchers continue to answer questions and problems. We believe that there is not any group working in morphology as we do. We show aspects that were not shown before. We think that it is a contribution. We do not intend to "solve a scientific problem." We intend to show details of the phenomenon, which were not shown before; it is also Science.

  1. In the results section, the author did not fully describe the results of each section, only briefly listed them.

Answer: We followed the journal instructions. The results section is just for a dry description of the findings. Anyway, we added more descriptions.

  1. The author should carefully consider what kind of scientific problem or phenomenon they want to illustrate through these experiments. The entire manuscript lacks logic, as if the author randomly listed some experimental results.

Answer: We work with Trichomonas morphology, and this subject is probably different from your research. Someone has to contribute with morphological data, which is also important for comprehending several parasite behaviors.

  1. In fact, this manuscript only describes the morphology of T. vaginalis in different states. It is difficult to determine whether there is mutual communication between T. vaginalis through these electron microscopy photos. Figure 12 and 13 can only illustrate the localization of E-cadherin protein in trophoblasts, as fluorescence staining of the cell membrane and cytoskeleton was not performed, making it impossible to determine whether E-cadherin is at the junction between cells. The experiment in Figure 14 lacks rigor, and its negative or positive results cannot determine whether there is communication between trophozoites. These fluorescent dyes may flow out of the trophozoite and exist in the culture medium, which is then randomly ingested by other trophozoites.

Answer: Trophoblasts? We did not use trophoblasts.

We do not agree with your statement that "these fluorescent dyes may flow out of the trophozoite and exist in the culture medium, which is then randomly ingested by other trophozoites".

We believe that the referee is not familiar with this kind of experiment. The fluorescent dye did not flow out, and in the case that it could occur, and in the case that the parasite ingested the dye, the dye would go to endosomes and lysosomes. Thus, fluorescent vesicles would be seen. Here, we tested plasma membrane labeling, searching for a possible mark in the region between two cells. Of course, affirmatives that there was no cell communication must be tested in different ways, and Science is made of several groups and experimental procedures. We present what we found. The experiment we made is known as Intracellular Dye Injection. In this experiment, a micropipette loaded with the dye Lucifer Yellow manages to penetrate the cell's plasma membrane simultaneously while its sealing occurs. After this procedure, the dye is injected through a hyperpolarizing pulse that repels the dye (with a negative charge) from inside the micropipette into the cell's cytoplasm.

Concerning the Fig. 14 (now another figure number), notice that the Lucifer Yellow dye is retained in the cell, considering the micropipette (filled with dye) that is present in the micrograph with the cell at its tip. This micrograph is important because it demonstrates that the dye is retained only in the injected cell, and the glow found around the cell is associated with the high intensity of glow caused by the dye, which can also be observed around the pipette. If the dye is internalized by the trophozoites randomly, it would be possible to see this dye delimited by the plasma membrane of the trophozoites, which does not occur in Figure 14 (letter c). Note that we can see the clear retention of the dye only in the injected cell. It is important to see that the micropipette is inside the injected cell, which demonstrates the same fluorescence intensity inside the pipette and the cell.

Therefore, given these findings, it is possible to state that there was no passage of dye into the cells adjacent to the cell that was injected with the Lucifer Yellow dye. Thus, intercellular communication is not formed through gap junctions in these cells.

Of course, other papers can reinforce or not our results, as in any Science field.

We performed a new experiment using a plasma membrane marker and presented in a new figure.

  1. There are too many images in this manuscript. It is recommended to merge them into less than 5 images.

Answer: We are saddened that the referee requested this. Our group works with morphology and likes to present a few images in each figure to facilitate reader understanding. Not to say that we didn't accept the referee's suggestion, we removed several images and concentrated on a few figures.

  1. Suggest changing this manuscript into a short communication for publication.

Answer: This manuscript is part of an invitation for a special issue on Trichomonas vaginalis in the journal Pathogens. When invited, I had to submit a summary of the proposal as a full article, and the editor accepted it. Furthermore, we are not going to switch to short communication.

  1. The language of this manuscript needs to be improved. 

Answer: We paid for English editing, and we have the certificate. However, few corrections were made, showing that spending money was unnecessary to edit the manuscript.

Minor problem

  1. At line 143, "9.109cells/ml" should be modified to "9x109cells/ml".
  2. At line 164, inconsistent spacing between lines.
  3. At line 166, font inconsistency.

Answer: Thanks, they were corrected.

  1. At line 189, “(+ -)55° with 2° increments)” ? Half bracket, centigrade(˚C).

Answer: The referee does not seem to know our method. The degrees we are referring to have nothing to do with C. It is not temperature but rather the tilt of the grid inside the transmission microscope.

  1. “interdigitations” or “inter digitations”.

Answer: We did not understand the comment since all the words "interdigitations" are written correctly.

  1. At line 166, text format.

Answer: OK, it was corrected.

  1. There are two conclusions in the manuscript.

Answer: Thanks, it was changed

Reviewer 3 Report

Comments and Suggestions for Authors

The manuscript entitled: “T. vaginalis: Monolayer and cluster formation - ultrastructural aspects using HR-SEM” addresses an interesting topic.

Infectious diseases, especially parasitic infections such as trichomoniasis, are one of the most common health issues and are considered as neglected diseases. As mentioned in this article, Trichomonas vaginalis causes human trichomoniasis and is a sexually transmitted infection (STI) that affects approximately 200 million people worldwide.

In fact, in the last century most attention has been drawn to cancer and autoimmune diseases and most studies have been done in this regard. Therefore, the importance of infectious diseases seems more than ever. Among these things that should be given more attention are diagnosis of the infection, treatment and drug resistance of the parasite, and host-parasite interaction especially the cytopathic effect of the parasite on the host cells.

Therefore such studies are necessary and important, which has been well addressed in the present study. The study is well designed and presented. All in all, on my view point, the study has a merit to be published.

Author Response

Reviewer 3

Thank you very much for your comments and attention to our manuscript.

Reviewer 4 Report

Comments and Suggestions for Authors

Trichomonas vaginalis is the most common parasite of STD worldwide distribution and infects more than 200 million population. The authors wanted to know the phenomenon of this parasite adhesion to inert substrates and the cluster formation as well as the details of the contact regions between the membranes of this parasite monolayers and clusters using methods including high resolution SEM, TEM, TEM tomography and some relatives. They found that the formation of a monolayer of tightly adherent cells when the parasites are in contact with an inert material and they demonstrated that no fusion was found between neighbor-attached cells. This work provide valuable data with high resolution EM photography to better understand the cluster formation and potential function of the cluster in vitro.

Special comments:

1. Introduction: This part is too weak regarding the background of this field. It would be greatly benefited for the readers if more details regarding the cluster, cytopathic effect of this parasite on the host cells could be mentioned. Authors mentioned that the virulent parasites change to an ameboid shape when in contact with target cells. We do have the avirulent strains of this parasite? What is the virulent parasite with what characteristics for the cells and for the animal model such mice? Is any relationship between virulence and the cluster formation in this parasite?

2. M&M: Based on our experience on the preparation for TEM and SEM materials (samples), using scraper to detach the monolayers frequently destroyed the structure of cell connection and lose details. Why not replace the plate with the ACLAR which could be embed in resin directly that does no need to detach cells.

3. Results: line 345-349, Did the injected cell still alive? How to decide “2 min” was the propriety time? Does it enough to reflect cell connection? line 364, “Scanning electron microscopy of SEM” should be modified to Scanning electron microscopy (SEM) shows….

4. Discussion: This part covers a lot but lacks of logicality and consistency, please refine the point and make this part succinct. Based on the observations in this work, authors should focus to discuss the potential formation mechanism and the function of the cluster when parasites attach to the target cell and the surface of the culture dish. I don’t think it is a good time to make a conclusion that no molecules communication (passage of molecules) between neighbor-attached cells (also see line 26 in the abstract) occurred since all the method used in this work may not detect the related molecules.

5. Some writing details should be check carefully. The unified of “ml” and “mL”.
Line 475, is it “Ca2 ” or “Ca2+ ”?

Author Response

Reviewer 4

  1. Introduction: This part is too weak regarding the background of this field. It would be greatly benefited for the readers if more details regarding the cluster, cytopathic effect of this parasite on the host cells could be mentioned. Authors mentioned that the virulent parasites change to an ameboid shape when in contact with target cells. We do have the avirulent strains of this parasite? What is the virulent parasite with what characteristics for the cells and for the animal model such mice? Is any relationship between virulence and the cluster formation in this parasite?

Answer: We changed several sentences in the manuscript and added new references concerning mice experiments.

  1. M&M: Based on our experience on the preparation for TEM and SEM materials (samples), using scraper to detach the monolayers frequently destroyed the structure of cell connection and lose details. Why not replace the plate with the ACLAR which could be embed in resin directly that does no need to detach cells.

Answer: We agree with the referee. Aclar is very good. However, we used a very careful way to detach the cells and observed at transmission electron microscopy that the cells we used in this study were morphologically perfect. Anyway, we are aware of your comment.

Concerning the protocol for SEM, the cells were grown directly on coverslips and processed in situ.No scrapers were used.

  1. Results: line 345-349, Did the injected cell still alive? How to decide "2 min" was the propriety time? Does it enough to reflect cell connection?

Answer: We changed and added some points. One of the authors has used this protocol for a long time, and has been published before. The cells were still alive since the flagella were vigorously beating.

Concerning if “Does it enough to reflect cell connection?”, we agree with the referee and have added comments about this in the manuscript.

Line 364, "Scanning electron microscopy of SEM," should be modified to Scanning electron microscopy (SEM) shows….

Answer: OK, it was changed.

  1. Discussion: This part covers a lot but lacks of logicality and consistency, please refine the point and make this part succinct. Based on the observations in this work, authors should focus to discuss the potential formation mechanism and the function of the cluster when parasites attach to the target cell and the surface of the culture dish. I don't think it is a good time to make a conclusion that no molecules communication (passage of molecules) between neighbor-attached cells (also see line 26 in the abstract) occurred since all the method used in this work may not detect the related molecules.

Answer: We agree with the referee and have changed several sentences in the manuscript.

  1. Some writing details should be check carefully. The unified of "ml" and "mL".
    Line 475, is it "Ca2" or "Ca2+"?

Answer: Thanks. We corrected.